# The Significant Association between Health Examination Results and Population Health: A Cross-Sectional Ecological Study Using a Nation-Wide Health Checkup Database in Japan

**DOI:** 10.3390/ijerph18020836

**Published:** 2021-01-19

**Authors:** Yukinori Nagakura, Hideaki Kato, Satoshi Asano, Yasuhiro Jinno, Shigeharu Tanei

**Affiliations:** 1School of Pharmacy at Fukuoka, International University of Health and Welfare, 137-1 Enokizu, Okawa 831-8501, Fukuoka, Japan; y-jinno@iuhw.ac.jp; 2School of Pharmacy, International University of Health and Welfare, 2600-1 Kitakanemaru, Ohtawara 324-8501, Tochigi, Japan; kato-h@iuhw.ac.jp (H.K.); satoshi.asano@iuhw.ac.jp (S.A.); 3Faculty of Pharmaceutical Sciences, Nihon Pharmaceutical University, 10281 Komuro, Ina-machi, Kitaadachi-gun, Saitama 362-0806, Japan; s-tanei@nichiyaku.ac.jp

**Keywords:** National Database of Health Insurance Claims and Specific Health Checkups of Japan (NDB) Open Data Japan, health checkup results, life expectancy, healthy life expectancy, population health, health habit improvement

## Abstract

In Japan, population health with life expectancy (LE) and healthy life expectancy (HALE) as indicators varies across the 47 prefectures (administrative regions). This study investigates how health examination results, including attitude toward improving life habits, are associated with population health. The association between health checkup variables and summary population health outcomes (i.e., life expectancy and healthy life expectancy) was investigated using a cross-sectional ecological design with prefectures as the unit of analysis. The medical records, aggregated by prefecture, gender, and age in the National Database of Health Insurance Claims and Specific Health Checkups of Japan (NDB) Open Data Japan, were used as health checkup variables. Body weight, blood pressure, liver enzymes, drinking habits, smoking habits, diabetes, serum lipids, and answers to questions regarding attitude toward improving health habits were significantly correlated to population health outcomes. Multiple regression analysis also revealed significant influence of these variables on population health. This study highlights that health examination results, including attitude toward improving health habits, are positively associated with population health. Consequently, implementing measures to improve health habits in response to the examination results could help the population maintain a healthy life.

## 1. Introduction

Population health significantly varies based on geography. Japan consists of 47 administrative regions called prefectures. Each prefecture has its unique natural features and culture. While health habits (e.g., diet, smoking, and drinking) are substantially associated with population health [1,2,3], these differ across prefectures. Life expectancy (LE), a commonly used summary population health indicator, varies significantly across prefectures [4]. Differences in LE among prefectures are increasing [5,6].

The Japanese health care system is attributed to the Bismarck-type health insurance where all citizens are covered under one of the public health insurance programs. The programs for individuals aged 40 to 74 years are divided into 2 categories, i.e., the employment-based health insurance system wherein company employees and their family members are enrolled and the residence-based National Health Insurance system which is for people not eligible for the employment-based insurance system [7]. Public health insurers are obliged to provide a specific health checkup for the insured individuals aged 40 to 74 years to tackle lifestyle diseases as per government policy [8]. In the 2016 fiscal year, more than half of the eligible adults participated in the checkup. The Ministry of Health, Labor, and Welfare of Japan (MHLW) constructed a nationwide database of the Japanese healthcare system. Based on this database, the MHLW publishes “the National Database of Health Insurance Claims and Specific Health Checkups of Japan (NDB) Open Data Japan”. This includes nationwide medical records. The fourth NDB Open Data Japan (the most recent publication) contains approximately 27.8 million medical records accumulated in 2016 and aggregated by prefecture, gender, and age [9]. It is an open publication accessible to the general public, including researchers.

LE, a measure of mortality, is commonly used as a summary measure to capture a “snapshot” of population health. The MHLW has surveyed and published LE, which is the average number of years that a newborn is expected to live, by prefecture, gender, and age every five years since 1965 [10]. Healthy life expectancy (HALE), an integrative measure which reflects mortality and morbidity and represents the overall level of population health [11], is also a similar summary measure [12]. The MHLW, since 2010, published HALE, the average length of time spent without limitation in daily activities, by prefecture, gender, and age every three years [13].

This study investigates how the health examination results, including attitude toward improving life habits, are associated with population health outcomes. The association between health checkup variables (i.e., laboratory tests and questions about health habits) in the fourth NDB Open Data Japan and summary population health outcomes (i.e., LE and HALE) was investigated using correlation study and multiple regression analysis across 47 Japanese prefectures wherein prefectures served as the unit of analysis.

## 2. Materials and Methods

### 2.1. Study Design

This study investigated the association between health checkup variables and summary population health outcomes (i.e., LE and HALE) using a cross-sectional ecological design with 47 Japanese prefectures wherein prefectures served as the unit of analysis. We analyzed the NDB Open Data Japan, which contains approximately 27.8 million medical records of adults nationwide aged 40 to 74 years in the 2016 fiscal year [9]. The Research Ethics Committee of the International University of Health and Welfare waived approval for this study.

### 2.2. Health Checkup Variables

This study used medical records aggregated by prefecture, gender, and age in the fourth NDB Open Data Japan [9] as health checkup variables. These were divided into two groups: laboratory tests and answers to questions regarding health habits. Specifically, the prefecture level average values in laboratory tests and percentage of respondents who selected each answer option in the questions were used as health checkup variables in the correlation analysis. The health checkup program included the following laboratory tests: body mass index (BMI), abdominal circumscript, fasting plasma glucose, HbA1c, systolic blood pressure, diastolic blood pressure, triglyceride, high density lipoprotein cholesterol (HDL-C), low density lipoprotein cholesterol (LDL-C), aspartate transaminase (AST), also known as glutamic oxaloacetic transaminase (GOT); alanine transaminase (ALT), also known as glutamate-pyruvate transaminase (GPT); γ-glutamyltransferase (γ-GT or γ-GTP), and hemoglobin. Table 1 summarizes laboratory tests and nationwide average value in each test [9].

Table 2 summarizes questions about health habits, answer options for each question, and the nationwide average percentages of respondents who selected each answer [9].

### 2.3. Standardization of Health Checkup Data

The health checkup raw data were standardized by age using the following formula: Standardized data = (∑ age-specific raw data in a 5-year age group × standard population in that age group)/(total population in standard population); the Japanese population in 2015 [14] was used as the standard population.

### 2.4. Summary Population Health Outcomes

The MHLW has published LE by prefecture and gender every five years since 1965. Table 3 shows the results of LE survey conducted in 2015 [10], and this was used as a summary population health variable. The MHLW has also published HALE, defined as the average duration of time spent without limitation in daily activities, by prefecture and gender every three years since 2010. Table 3 indicates the results of HALE survey conducted in 2016, and this was used as a summary population health variable [13]. Due to the massive earthquake in the Kumamoto prefecture in 2016, there were no HALE data for this location. Thus, we conducted a correlation analysis between health checkup variables and HALE for the remaining 46 prefectures.

### 2.5. Statistical Analysis

We evaluated the association between health checkup variables and summary population health outcomes (i.e., LE and HALE) by gender with the 47 prefectures where prefectures served as the unit of analysis. Correlations between the two variables were examined using *t*-test and Pearson’s correlation coefficient (*r*) for each pair of variables. Additionally, a multiple regression analysis was performed using a model with each summary population health outcome as the objective variable and two groups of health checkup variables (i.e., the laboratory tests and the answer to the questions regarding health habits) as candidate explanatory variables. The backward stepwise selection procedure was used to select the explanatory variables from the candidate variables for final inclusion in the regression model with the criteria set at *p* < 0.2. In the regression analysis using the laboratory tests, BMI, fasting plasma glucose, HbA1c, systolic blood pressure, triglyceride, HDL-C, LDL-C, γ-GT, and hemoglobin were included as candidate explanatory variables (BMI was used as body weight-related variable among BMI and abdominal circumscript. Systolic blood pressure was used as the blood pressure-related variable among systolic and diastolic blood pressure. γ-GT was used as liver enzyme-related variable among GOT, GPT, and γ-GT). In the regression analysis using the answer to the questions regarding health habits, all question items (Q1–22) were included as candidate explanatory variables. Only answer option 1 was the target of analysis for question items with 3 or more answering options. A quantile–quantile (Q–Q) plot was used to verify the normality assumption of the residuals in the regression analysis. The significance of individual regression coefficients was tested by *t*-test. *p* values < 0.05 were considered statistically significant. Statistical analysis was performed with BellCurve for Excel version 3.20 (Social Survey Research Information Co., Ltd., Tokyo, Japan).

## 3. Results

### 3.1. Correlation Analysis Using Laboratory Tests as Health Checkup Variables

Table 4 summarizes the results of correlation analysis between laboratory tests as health checkup variables and summary population health outcomes. A positive correlation implying statistically significant linear relation was found between the following pairs of variables: (1) HbA1c—HALE in females and (2) LDL-C—LE in males. Meanwhile, a negative correlation implying statistically significant linear relation was present between the following pairs of variables: (1) BMI–LE in both genders, (2) systolic blood pressure–LE in both genders, (3) GOT–LE in both genders and HALE in males, (4) GPT–LE in both genders and HALE in males, (5) γ-GT–LE in both genders and HALE in males, (6) diastolic blood pressure–LE in both genders, (7) fasting plasma glucose–LE in males, (8) triglyceride–LE in males, (9) HDL-C–HALE in females.

### 3.2. Multiple Regression Analysis Using Laboratory Tests as Explanatory Variables

Multiple regression analysis for predicting population health outcomes using laboratory tests as explanatory variables was performed. Table 5 summarizes the results by gender. BMI, systolic blood pressure, and γ-GT were significantly associated negatively with LE in males (*R*^2^ = 0.5725, *F* = 19.1987, *p* < 0.001). HbA1c positively and γ-GT negatively showed significant associations with HALE in males (*R*^2^ = 0.3001, *F* = 9.2195, *p* < 0.001). BMI, systolic blood pressure, and HDL-C showed a statistically significant negative association with LE in females (0.2964, *F* = 6.0374, *p* = 0.0016). γ-GT was significantly associated negatively with HALE in females (*R*^2^ = 0.2959, *F* = 3.3614, *p* = 0.0125). Normality of residual distribution in each regression analysis was met as assessed by Q–Q plot.

### 3.3. Correlation Analysis Using Questions about Health Habits as Health Checkup Variables

Table 6 summarizes the result of correlation analysis between results of questions about health habits as health checkup variables and summary population health outcomes. A positive correlation implying statistically significant linear relation was found between the following pairs of variables: (1) Question (Q)2/Yes (Y) (hypoglycemic drug)–HALE in females, (2) Q6/Y (chronic kidney failure)–HALE in females, (3) Q7/Y (anemic)–LE in males, (4) Q18/answer (A)3 (drink rarely)–LE in males and HALE in females, (5) Q19/A1 (drink less than 180 mL of sake)–LE in males, (6) Q21/A1 (no intention of improving health habits)–HALE in females, and (7) Q21/A5 (already working on improving health habits for 6 months or more)–LE in both genders. Meanwhile, a negative correlation implying statistically significant linear relation was present between the following pairs of variables: (1) Q1/Y (antihypertensive drug)–LE in both genders, (2) Q2/Y (hypoglycemic drug)–LE in males, (3) Q8/Y (habitual smoking)–LE in both genders, (4) Q9/Y (body weight gain)–HALE in males, (5) Q13/Y (body weight fluctuation)–LE in females and HALE in males, (6) Q14/A1 (fast eating speed)–HALE in males, (7) Q18/A1 (drink every day)–LE in males and HALE in both genders, (8) Q18/A2 (drink sometimes)–HALE in females, (9) Q19/A3 (drink 360–540 mL of sake)–LE in males, (10) Q19/A4 (drink more than 540 mL of sake)–LE in males, (11) Q21/A2 (improvement in health habits within 6 months from now)–HALE in both genders, and (12) Q22/Y (intention to take health instructions)–HALE in both genders.

### 3.4. Multiple Regression Analysis Using Questions about Health Habits as Explanatory Variables

Multiple regression analysis for predicting population health outcomes using questions about health habits as explanatory variables was performed. Table 7 summarizes results by gender. Q1/Y and Q8/Y were factors significantly associated with LE in males (*R*^2^ = 0.5273, *F* = 9.1472, *p* < 0.001). Q4/Y, Q5/Y, Q8/Y, Q13/Y, Q15/Y, Q18/A1, and Q22/Y were factors significantly associated with HALE in males (*R*^2^ = 0.7216, *F* = 9.0702, *p* < 0.001). Q8/Y, Q9/Y, and Q17/Y were factors significantly associated with LE in females (*R*^2^ = 0.5798, *F* = 4.9678, *p* < 0.001). Q2/Y, Q13/Y, Q18/A1, and Q20/Y were factors significantly associated with HALE in females (*R*^2^ = 0.5701, *F* = 7.1982, *p* < 0.001). Normality of residual distribution in each regression analysis was met as assessed by Q–Q plot.

## 4. Discussion

This study was a cross-sectional ecological analysis with prefectures as the unit of analysis using a nationwide health checkup database in Japan conducted to demonstrate that health examination results are associated with summary population health.

*Body weight-related variables:* BMI and LE were negatively associated in both genders in the correlation study and regression analysis. A previous study suggested that the association between BMI and relative mortality risk was J-shaped. This implied that high and extremely low BMI values are associated with increased all-cause mortality. The increased mortality in those with extremely low BMI was partly due to the inclusion of individuals with diseases that cause weight loss and premature death and residual confounding by smoking [15]. The linear negative relation between BMI and LE in this study could happen, probably because individuals with diseases causing weight loss and premature death tend to not participate in health checkups.

The percentage for Q13/Y (Have you experienced body weight fluctuation of ±3 kg or more during the last year?/Yes) was negatively correlated to LE in females and HALE in males. Its negative influence on HALE was also demonstrated by the regression analysis in both genders. Relapse of weight gain often occurs after dieting, and such weight fluctuation is associated with diseases, including cardiovascular diseases [16]. A recent meta-analysis concluded that weight fluctuation is associated with an elevation of all-cause mortality risk [17]. Additionally, the percentage for Q14/Y (Is your eating speed faster than others?/Yes) was negatively correlated to HALE in males. This result was expected based on the assumption that speedy eating is associated with prevalence of obesity and diabetes [18,19]. This in turn increases the risk of cardiovascular diseases, one of primary causes of morbidity and mortality.

*Blood pressure-related variables:* Elevated blood pressure was substantially associated negatively with LE in both genders in the correlation study and regression analysis. This was consistent with existing evidence that high blood pressure is a major risk for mortality and morbidity [20,21]. Hypertension is associated with a wide range of acute and chronic cardiovascular diseases, such as angina and heart failure [22]. The percentage for Q1/Y (Are you currently under any medication for high blood pressure?/Yes) was negatively associated with LE in both genders in the correlation study and in males in the regression study. Since those participants who selected “Yes” presumably suffer from hypertension, this negative correlation could be due to the negative impact of hypertension on LE [20,21].

*Liver enzymes-related variables:* Liver enzymes were negatively correlated to LE in both genders and with HALE in females. A negative impact of γ-GT on LE and HALE in males and HALE in females was also demonstrated in the regression analysis. This negative association was expected in light of existing evidence that the elevation of these liver enzymes is a sensitive marker of various liver diseases, including alcohol- and drug-induced liver injury, hepatitis, hereditary hemochromatosis, and cirrhosis [23]. Consequently, high levels of these enzymes are predictors of all-cause mortality among the elderly [24] and the general population [25,26].

*Drinking habit-related variables:* The percentage for Q18/A1 (How often do you drink alcohol (e.g., sake, shochu, beer, whisky, wine)?/Every day) was negatively correlated to LE in males and HALE in both genders. Its negative influence on HALE on both genders was also demonstrated by the regression analysis. The percentage for Q19/A3 (How much do you drink in terms of sake per day?/360–540 mL of sake) was negatively correlated to LE in males. Conversely, the percentage that selected A1 (less than 180 mL of sake) was positively correlated to LE in males. Further, the percentage for Q18/A3 (rarely drink) was positively correlated to LE in males and HALE in females. These results are consistent with the existing evidence that habitual heavy drinking is associated with high mortality and morbidity, although light drinking may reduce the risk of some cardiovascular diseases. Indeed, the relationship between alcohol consumption and mortality risk is generally J-shaped, i.e., the mortality risk is reduced by light consumption compared to abstinence but increases steeply as consumption increases [27,28]. Excessive alcohol consumption is one of the leading causes of premature mortality [29], and there is a strong association between heavy drinking and various diseases, such as cancer, cardiovascular disease, liver disease, and diabetes [30].

*Smoking habit-related variables:* The percentage for Q8/Y (Are you a habitual cigarette smoker (defined as a person who smoked a total of over 100 cigarettes or for over 6 months and has smoked in the last month) at present?/Yes) is negatively correlated to LE in both genders. Its negative influence on LE in both genders was also proven by the regression analysis. This was in accordance with the accumulated evidence that a smoking habit is associated with high risk of various chronic diseases (e.g., cancers and cardiovascular diseases) and decrease of LE [31,32,33,34]. On the other hand, the correlation with HALE was insignificant in this study. This result was inconsistent with previous studies which showed that smoking habit is associated with reduction of HALE [35,36,37]. This inconsistency could be due to different study designs. For instance, subjects aged 65 years or older were targeted in the previous study [35], while those aged 40–74 years were included in this study.

*Diabetes-related variables:* Fasting blood glucose level was negatively correlated to LE in males. This was consistent with the evidence that diabetes is associated with premature death caused by various diseases, such as cardiovascular diseases, cancers, and infectious diseases [38]. HbA1c exhibited a positive correlation with HALE in females. The relation between HbA1c and all-cause mortality in non-diabetic examinees was reportedly reverse J-shaped with an HbA1c of 5.4% as the lowest mortality risk. All-cause mortality risk does not increase significantly above an HbA1c level of 5.4% for non-diabetic examinees, although the risk is significantly higher in the low range, i.e., less than 5.0% [39]. Given that most participants were non-diabetic individuals, the result seemed consistent with this literature.

The percentage for Q2/Y (Do you take insulin injections or other medications to reduce blood glucose level at present?/Yes) was negatively correlated to LE in males. Since the participants who selected “Yes” presumably suffer diabetes, this negative correlation was due to the negative impact of diabetes on population health outcomes [40,41]. Meanwhile, the positive association between the percentage for Q2/Y and HALE in females was demonstrated by the correlation study and regression analysis. One possible interpretation for the paradoxical result would be that the participants who selected “Yes” take measures to control blood glucose and prevent diabetic complications, such as retinopathy, neuropathy, and nephropathy. Such preventive interventions (including hypoglycemic drugs) prevent or delay disabilities caused by diabetic complications because a previous study [42] established that adoption of intensive diabetes management delays or prevents serious diabetic complications.

*Serum lipids-related variables:* Triglyceride was negatively associated with LE in males, in line with the reported negative impact of triglyceride on health. High triglyceride seems to be associated with the high mortality and morbidity of cardiovascular diseases and cancer based on epidemiological and genetic evidence [43]. Excessive accumulation of triglyceride in somatic cells is involved in pathophysiology for obesity [44], which is, in turn, associated with mortality and various comorbidities including diabetes, nonalcoholic fatty liver disease, and cardiomyopathy [45].

LDL-C and LE were positively correlated in males in the correlation study. Additionally, a negative association between HDL-C and HALE in females was observed in the correlation study and regression analysis. These results were inconsistent with their roles in the pathophysiology of atherosclerosis. Currently, LDL-C is considered a major causal factor for cholesterol transport to atherosclerotic lesions, whereas HDL-C performs a reverse transport of cholesterol to the liver [46]. Elevated LDL-C and decreased HDL-C levels are considered risk factors for atherosclerosis in cardiovascular disease [47]. There are also alternative observations to which this study’s results seem proximate. A study demonstrated a significant trend wherein LDL-C is negatively associated with all-cause mortality, i.e., low LDL-C is associated with high mortality risk [48,49,50]. Regarding HDL-C, a meta-analysis suggested that high HDL-C does not reduce the risk of cardiovascular diseases [51]. According to a recent study, the association between HDL-C level and all-cause mortality is U-shaped, and extremely high and low HDL-C is associated with high risk of all-cause mortality [52]. One possible interpretation would be that the participants who showed high LDL-C or low HDL-C level could make an early start of dyslipidemia treatments, which could consequently help such populations maintain a healthy life.

*Variables related to attitude toward improving life habits:* Answers to questions about the attitude toward improving health habits were significantly associated with the population health. The percentage for Q21/A5 (Are you going to improve your life habits such as diet and exercise?/already working on health habit improvement for 6 months), i.e., the most positive attitude to improving health habits, was positively correlated to LE in both genders. Paradoxically, the percentage for Q21/A1 (no intention of improving health habits) was also positively correlated to HALE in females. Presumably, the participants who selected A1 do not need to improve health habits because they already adopted healthy habits. Consequently, they maintain a healthy life for a longer period. The percentage for Q21/A2 (health habit improvement within 6 months from now) was negatively correlated to HALE in both genders. The participants selecting A2 hesitated to practice improvement of health habits immediately despite recognizing the need to improve. They might be susceptible to developing diseases associated with disability.

The percentage for Q22/Y (Would you like to receive instructions on life habit improvements?/Yes) was negatively correlated to HALE in both genders. This negative impact on HALE was also indicated by the regression analysis. Given the assumption that the participants selecting “Yes” have concerns about their health and find it difficult to improve life habits on their own, they might be vulnerable to diseases associated with disabilities.

*Limitations:* This study employed a cross-sectional ecological study design and used the health checkup data and the summary population health data collected in 2016 and 2015, respectively. One of the limitations of this study is that the health checkup data were aggregated prefecture-wise from the perspective of privacy protection. Hence, individual level analysis could not be conducted, limiting the findings to only indicate that health examination results are significantly associated with population health at the prefecture level. A study that uses individual data remains to be conducted to determine the factors associated with individual health. The other limitation is related to the selection bias attributable to the relatively low participation rate (slightly higher than 50%) for the health checkup. Since participation is not mandatory for eligible adults (aged 40 to 74 years), the participation rate remains low and varies based on the insurers providing the checkup programs [53]. For example, participation rate was 75% for people insured by large companies’ insurance associations, one of Employee Health Insurance, and 37% for those insured by municipality-based National Health Insurance [53]. Thus, the participation rate is generally higher for employees than for self-employed persons, retirees, and non-working dependents. This difference of properties between participants and non-participants could affect the findings. Indeed, the properties in terms of various factors, such as socio-economic condition, education level, financial status, and mental condition could affect population health outcomes. For example, a recent study which performed multiple regression analysis using Japanese population data aggregated by prefecture demonstrated that LE at 65 years of age in females was significantly affected by healthcare resource factors (beds per capita, doctors per capita, and medical expenses for the elderly) and an environmental factor (air pollution) [54]. Factorial analyses by including and controlling multiple factors, such as socio-economic-, mental condition-, and environment-associated variables must be focused on by future research.

## 5. Conclusions

This study highlights that health examination results, including attitude toward improving health habits, are positively associated with population health at the prefecture level. Thus, implementing measures to improve health habits in response to the examination results could help the population maintain a healthy life.

## Figures and Tables

**Table 1 ijerph-18-00836-t001:** Health checkup variables (laboratory tests).

Laboratory Test (Unit)	Nation-Wide Average Value for Adults Aged 40–74	
	Male	Female
BMI (kg/m^2^)	24.0	22.3
Abdominal circumscript (cm)	85.0	79.6
Fasting plasma glucose (mg/dL)	101.1	93.6
HbA1c (%)	5.7	5.6
Systolic blood pressure (mmHg)	126.4	121.2
Diastolic blood pressure (mmHg)	78.6	72.6
Triglyceride (mg/dL)	133.6	94.4
HDL-C (mg/dL)	58.0	70.0
LDL-C (mg/dL)	123.2	124.3
GOT (AST) (U/L)	25.0	21.7
GPT (ALT) (U/L)	27.0	18.5
γ-GT (γ-GTP) (U/L)	51.1	25.5
Hemoglobin (g/dL)	15.1	13.0

**Table 2 ijerph-18-00836-t002:** Health checkup variables (questions on health habits).

No.	Question (Answer Options)	Ans.	Nation-Wide %	
			M	F
Q1	Are you currently under any medication for high blood pressure? (Y/N)	Yes	22.7	17.5
Q2	Do you take insulin injections or other medications to reduce blood glucose level at present? (Y/N)	Yes	7.2	3.2
Q3	Do you take medication to reduce cholesterol level at present? (Y/N)	Yes	12.1	15.1
Q4	Have you ever been diagnosed as stroke (e.g., cerebral hemorrhage, cerebral infarction) by a doctor or got treated for it? (Y/N)	Yes	3.0	1.4
Q5	Have you ever been diagnosed as heart disease (e.g., angina, myocardial infarction) by a doctor or got treated for it? (Y/N)	Yes	5.0	2.4
Q6	Have you ever been diagnosed with chronic kidney failure by a doctor or got treated (e.g., dialysis) for it? (Y/N)	Yes	1.4	0.3
Q7	Have you ever been diagnosed as anemic by a doctor? (Y/N)	Yes	4.0	19.0
Q8	Are you a habitual cigarette smoker (defined as a person who smoked a total of over 100 cigarettes or for over six months and has smoked in the last month) at present? (Y/N)	Yes	33.6	9.7
Q9	Have you gained 10 kg or more compared to your body weight when you were 20 years-old? (Y/N)	Yes	44.2	25.8
Q10	Have you been habitually doing slightly sweaty exercise (30 min or more per session and two days or more per week) for over a year? (Y/N)	Yes	29.1	26.4
Q11	Have you been walking or doing any equivalent amount of physical activity over an hour per day in everyday life? (Y/N)	Yes	39.4	43.2
Q12	Is your walking speed faster than the speed of those of almost the same age and of the same gender? (Y/N)	Yes	48.0	45.3
Q13	Have you experienced body weight fluctuation of ±3 kg or more during the last year? (Y/N)	Yes	25.2	21.1
Q14	Is your eating speed faster than others? (A1) Faster (A2) Ordinary (A3) Slower	A1	34.7	26.6
		A2	58.2	64.6
		A3	7.0	8.8
Q15	Do you have dinner within two hours before going to bed three times or more per week? (Y/N)	Yes	36.1	17.8
Q16	Do you have a bedtime snack after dinner three times or more per week? (Y/N)	Yes	13.9	17.2
Q17	Do you skip breakfast three times or more per week? (Y/N)	Yes	20.2	11.3
Q18	How often do you drink alcohol (e.g., sake, shochu, beer, whisky, wine)? (A1) Every day (A2) Sometimes (A3) Rarely or never	A1	39.9	13.1
		A2	30.5	27.2
		A3	29.5	59.7
Q19	How much do you drink in terms of sake per day? (180 mL of sake is equivalent to 500 mL of beer, 80 mL of shochu, 60 mL of whisky, 240 mL of wine) (A1) < 180 mL (A2) 180–360 mL (A3) 360–540 mL (A4) < 540 mL	A1	40.0	75.5
		A2	35.3	18.4
		A3	18.3	4.8
		A4	6.4	1.3
Q20	Are you getting enough rest with sleep? (Y/N)	Yes	64.6	63.2
Q21	Are you going to improve your life habits such as diet and exercise? (A1) I don’t intend to improve life habits. (A2) I will improve life habits in approximately six months. (A3) I will improve life habits in approximately 1 month and have already started little by little. (A4) I have already been working on the improvement for less than six months. (A5) I have already been working on the improvement for six months or more.	A1	29.8	24.6
		A2	32.2	35.6
		A3	13.2	15.7
		A4	8.6	9.2
		A5	16.2	14.9
Q22	Would you like to receive instructions on life habit improvements? (Y/N)	Yes	35.3	40.4

(): Answer options in each question, Ans.: answer selected.; M: males; F: females; Nation-wide %: nation-wide average percentages of respondents (adults aged 40–74) selecting the answer.

**Table 3 ijerph-18-00836-t003:** Summary population variables (LE and HALE by prefecture and gender).

Prefecture	LE ^1^		HALE ^2^		Prefecture	LE ^1^		HALE ^2^	
	Male	Female	Male	Female		Male	Female	Male	Female
Hokkaido	80.28	86.77	71.98	73.77	Shiga	81.78	87.57	72.30	74.07
Aomori	78.67	85.93	71.64	75.14	Kyoto	81.40	87.35	71.85	73.97
Iwate	79.86	86.44	71.85	74.46	Osaka	80.23	86.73	71.50	74.46
Miyagi	80.99	87.16	72.39	74.43	Hyogo	80.92	87.07	72.08	74.23
Akita	79.51	86.38	71.21	74.53	Nara	81.36	87.25	71.39	74.10
Yamagata	80.52	86.96	72.61	75.06	Wakayama	79.94	86.47	71.36	74.42
Fukushima	80.12	86.40	71.54	75.05	Tottori	80.17	87.27	71.69	74.14
Ibaraki	80.28	86.33	72.50	75.52	Shimane	80.79	87.64	71.71	75.74
Tochigi	80.10	86.24	72.12	75.73	Okayama	81.03	87.67	71.54	75.09
Gunma	80.61	86.84	72.07	75.20	Hiroshima	81.08	87.33	71.97	73.62
Saitama	80.82	86.66	73.10	74.67	Yamaguchi	80.51	86.88	72.18	75.18
Chiba	80.96	86.91	72.37	75.17	Tokushima	80.32	86.66	71.34	74.04
Tokyo	81.07	87.26	72.00	74.24	Kagawa	80.85	87.21	72.37	74.83
Kanagawa	81.32	87.24	72.30	74.63	Ehime	80.16	86.82	71.33	74.59
Niigata	80.69	87.32	72.45	75.44	Kochi	80.26	87.01	71.37	75.17
Toyama	80.61	87.42	72.58	75.77	Fukuoka	80.66	87.14	71.49	74.66
Ishikawa	81.04	87.28	72.67	75.18	Saga	80.65	87.12	71.60	75.07
Fukui	81.27	87.54	72.45	75.26	Nagasaki	80.38	86.97	71.83	74.71
Yamanashi	80.85	87.22	73.21	76.22	Kumamoto	81.22	87.49	- ^3^	- ^3^
Nagano	81.75	87.67	72.11	74.72	Oita	81.08	87.31	71.54	75.38
Gifu	81.00	86.82	72.89	75.65	Miyazaki	80.34	87.12	72.05	74.93
Shizuoka	80.95	87.10	72.63	75.37	Kagoshima	80.02	86.78	72.31	75.51
Aichi	81.10	86.86	73.06	76.32	Okinawa	80.27	87.44	71.98	75.46
Mie	80.86	86.99	71.79	76.30					

^1^ LE in 2015 published by the MHLW [10]; ^2^ HALE in 2016 published by the MHLW [13]; ^3^ HALE survey in 2016 did not include Kumamoto prefecture due to the massive earthquake which struck this area in 2016. LE: life expectancy; HALE: healthy life expectancy.

**Table 4 ijerph-18-00836-t004:** Correlation coefficient between health check-up variables (laboratory tests) and summary population health outcomes (LE and HALE).

Laboratory Tests	LE				HALE			
	Male		Female		Male		Female	
	*r*	*p*	*r*	*p*	*r*	*p*	*r*	*p*
BMI (kg/m^2^)	−0.460	0.001	−0.378	0.009	−0.215	0.150	0.096	0.525
AC (cm)	−0.170	0.253	−0.048	0.749	−0.247	0.098	0.248	0.096
FPG (mg/dL)	−0.460	0.001	−0.069	0.646	−0.245	0.101	0.257	0.085
HbA1c (%)	0.091	0.541	0.198	0.183	0.290	0.051	0.408	0.005
Systolic BP (mmHg)	−0.497	<0.001	−0.403	0.005	−0.287	0.053	0.010	0.948
Diastolic BP (mmHg)	−0.359	0.013	−0.412	0.004	−0.065	0.669	0.057	0.709
Triglyceride (mg/dL)	−0.478	<0.001	−0.058	0.697	−0.267	0.073	0.019	0.903
HDL-C (mg/dL)	0.043	0.772	0.112	0.454	−0.164	0.277	−0.303	0.041
LDL-C (mg/dL)	0.344	0.018	0.090	0.549	0.096	0.524	−0.241	0.106
GOT (AST) (U/L)	−0.664	<0.001	−0.442	0.002	−0.426	0.003	−0.166	0.270
GPT (ALT) (U/L)	−0.637	<0.001	−0.351	0.016	−0.329	0.026	−0.051	0.738
γ-GT (γ-GTP) (U/L)	−0.671	<0.001	−0.304	0.038	−0.469	0.001	−0.228	0.127
Hemoglobin (g/dL)	−0.199	0.179	−0.112	0.454	0.076	0.614	−0.079	0.601

*r*: Pearson’s correlation coefficient, AC: abdominal circumscript, FPG: fasting plasma glucose, BP: blood pressure. The cells filled with red and blue represent positive and negative correlations with statistically significant level (*p* < 0.05) by *t*-test, respectively.

**Table 5 ijerph-18-00836-t005:** Multiple regression analysis between population health outcomes and laboratory tests as explanatory variables.

Objective Variables	Explanatory Variables	B with 95% CI	*p*	β	VIF	*R* ^2^
LE in males	BMI	−0.5756 (−1.0755 to −0.0757)	0.0250 *	−0.2487	1.1542	0.5725
	Systolic BP	−0.1899 (−0.3349 to −0.0450)	0.0114 *	−0.2847	1.1674	
	γ-GT	−0.0871 (−0.1298 to −0.0444)	<0.001 **	−0.4722	1.3229	
HALE in males	HbA1c	3.6713 (0.3306 to 7.0120)	0.0320 *	0.2828	1.0002	0.3001
	γ-GT	−0.0756 (−0.1175 to −0.0338)	<0.001 **	−0.4648	1.0002	
LE in females	BMI	−0.6269 (−1.0734 to −0.1804)	0.0070 **	−0.4941	1.8610	0.2964
	Systolic BP	−0.1570 (−0.2759 to −0.0380)	0.0109 *	−0.3736	1.2041	
	HDL-C	−0.1160 (−0.2289 to −0.0031)	0.0442 *	−0.3681	1.9278	
HALE in females	FPG	0.1397 (−0.0227 to 0.3021)	0.0898	0.2715	1.3848	0.2959
	HbA1c	3.3961 (−1.4823 to 8.2746)	0.1672	0.2172	1.3540	
	Triglyceride	−0.0399 (−0.1003 to 0.0205)	0.1897	−0.2065	1.3610	
	HDL-C	−0.1095 (−0.2480 to 0.0290)	0.1179	−0.2602	1.5067	
	γ-GT	−0.2262 (−0.4243 to −0.0281)	0.0262 *	−0.3442	1.2635	

BMI, FPG, HbA1c, systolic BP, triglyceride, HDL-C, LDL-C, γ-GT, and hemoglobin were included as candidate explanatory variables. The explanatory variables, which were finally included in the regression model and presented in this table, were selected from the candidate variables using the backward stepwise selection procedure with the criteria of *p* < 0.2. CI: confidence interval; B: partial regression coefficient; β: standardized partial regression coefficient; VIF: variance inflation factor; *R*^2^: coefficient of determination; BP: blood pressure; FPG: fasting plasma glucose; * *p* < 0.05; ** *p* < 0.01 by *t*-test.

**Table 6 ijerph-18-00836-t006:** Correlation coefficient between health checkup variables (questions on health habits) and summary population health outcomes (LE and HALE).

Question		LE				HALE			
		Male		Female		Male		Female	
No.	Ans.	*R*	*p*	*r*	*p*	*r*	*p*	*r*	*p*
Q1	Yes	−0.611	<0.001	−0.395	0.006	0.143	0.342	0.264	0.076
Q2	Yes	−0.436	0.002	−0.237	0.108	−0.063	0.675	0.425	0.003
Q3	Yes	0.275	0.061	0.013	0.929	0.091	0.546	0.177	0.240
Q4	Yes	−0.235	0.111	0.031	0.837	−0.110	0.469	0.046	0.759
Q5	Yes	−0.201	0.176	−0.089	0.554	−0.120	0.428	−0.025	0.868
Q6	Yes	−0.070	0.639	0.172	0.247	0.124	0.410	0.379	0.009
Q7	Yes	0.337	0.020	0.229	0.122	0.023	0.881	−0.141	0.349
Q8	Yes	−0.473	<0.001	−0.472	<0.001	−0.099	0.514	−0.139	0.358
Q9	Yes	−0.074	0.623	−0.269	0.068	−0.298	0.044	0.061	0.686
Q10	Yes	0.033	0.825	−0.053	0.724	−0.156	0.300	−0.075	0.618
Q11	Yes	−0.116	0.439	−0.150	0.315	−0.003	0.986	0.034	0.824
Q12	Yes	0.060	0.690	−0.119	0.425	−0.134	0.373	−0.236	0.115
Q13	Yes	−0.060	0.689	−0.295	0.044	−0.414	0.004	−0.265	0.076
Q14	A1	0.119	0.424	0.095	0.523	−0.304	0.040	−0.054	0.721
	A2	−0.148	0.320	−0.209	0.159	0.266	0.074	0.203	0.177
	A3	0.084	0.573	0.208	0.161	−0.034	0.822	−0.239	0.110
Q15	Yes	0.118	0.428	−0.084	0.573	−0.009	0.953	−0.276	0.064
Q16	Yes	−0.031	0.837	0.090	0.549	−0.284	0.056	−0.258	0.084
Q17	Yes	−0.142	0.343	−0.030	0.841	−0.191	0.203	−0.084	0.579
Q18	A1	−0.327	0.025	−0.067	0.654	−0.343	0.020	−0.481	<0.001
	A2	0.052	0.730	0.079	0.597	0.239	0.109	−0.320	0.030
	A3	0.308	0.035	−0.020	0.892	0.124	0.413	0.434	0.003
Q19	A1	0.486	<0.001	0.195	0.190	0.137	0.362	0.196	0.191
	A2	−0.181	0.225	−0.215	0.146	−0.001	0.996	−0.186	0.217
	A3	−0.476	<0.001	−0.136	0.363	−0.113	0.456	−0.197	0.181
	A4	−0.348	0.017	−0.057	0.705	−0.206	0.169	−0.119	0.430
Q20	Yes	−0.228	0.124	−0.093	0.533	−0.146	0.334	0.220	0.142
Q21	A1	−0.107	0.473	−0.026	0.864	0.201	0.181	0.391	0.007
	A2	−0.190	0.200	−0.094	0.530	−0.428	0.003	−0.332	0.024
	A3	−0.022	0.885	−0.224	0.131	0.136	0.366	−0.057	0.709
	A4	0.166	0.264	0.225	0.128	−0.078	0.608	−0.136	0.366
	A5	0.347	0.017	0.386	0.007	0.237	0.113	0.042	0.782
Q22	Yes	−0.274	0.063	−0.062	0.677	−0.565	<0.001	−0.455	0.002

Ans.: answer selected; *r*: Pearson’s correlation coefficient. The cells filled with red and blue represent positive and negative correlations with statistically significant level (*p* < 0.05) by *t*-test, respectively.

**Table 7 ijerph-18-00836-t007:** Multiple regression analysis between population health and questions on health habits as explanatory variables.

Objective Variables	Explanatory Variables	B with 95% CI	*p*	β	VIF	*R* ^2^
LE in males	Q1/Y	−0.1886 (−0.2722 to −0.1051)	<0.001 **	−0.6107	1.5543	0.5273
	Q3/Y	0.0954 (−0.0225 to 0.2132)	0.1100	0.1785	1.0358	
	Q8/Y	−0.0702 (−0.1287 to −0.0117)	0.0198 *	−0.2989	1.3175	
	Q16/Y	−0.0509 (−0.1159 to 0.0140)	0.1210	−0.1773	1.0871	
	Q20/Y	0.0289 (−0.0131 to 0.0709)	0.1719	0.1851	1.5371	
HALE in males	Q3/Y	−0.0777 (−0.1760 to 0.0206)	0.1176	−0.1648	1.3261	0.7216
	Q4/Y	0.6377 (0.3139 to 0.9614)	<0.001 **	0.6690	3.5177	
	Q5/Y	−0.3094 (−0.5294 to −0.0895)	0.0072 **	−0.4134	2.6328	
	Q8/Y	0.0587 (0.0065 to 0.1108)	0.0285 *	0.2824	1.9203	
	Q9/Y	−0.0563 (−0.1243 to 0.0116)	0.1013	−0.2990	3.9668	
	Q13/Y	−0.1171 (−0.2159 to −0.0184)	0.0215 *	−0.3773	3.0870	
	Q15/Y	0.1066 (0.0576 to 0.1556)	<0.001 **	0.5309	1.8131	
	Q17/Y	−0.0550 (−0.1183 to 0.0084)	0.0869	−0.2547	2.6274	
	Q18/A1	−0.1008 (−0.1414 to −0.0602)	<0.001 **	−0.6913	2.3617	
	Q22/Y	−0.1061 (−0.1576 to −0.0547)	<0.001 **	−0.4628	1.5351	
LE in females	Q3/Y	0.0650 (−0.0208 to −0.1508)	0.1332	0.2024	1.4869	0.5798
	Q5/Y	−0.1591 (−0.3894 to −0.0712)	0.1698	−0.1900	1.5760	
	Q6/Y	0.2877 (−0.0659 to −0.6412)	0.1077	0.3439	3.7218	
	Q8/Y	−0.1418 (−0.2129 to −0.0707)	<0.001 **	−0.6363	2.1218	
	Q9/Y	−0.0710 (−0.1354 to −0.0065)	0.0318 *	−0.4844	4.0310	
	Q12/Y	0.0262 (−0.0039 to 0.0563)	0.0859	0.2789	2.1365	
	Q13/Y	−0.0809 (−0.1981 to −0.0362)	0.1698	−0.3324	4.8243	
	Q16/Y	0.0287 (−0.0090 to −0.0664)	0.1317	0.1941	1.3570	
	Q17/Y	0.1291 (0.0501 to −0.2081)	0.0021 **	0.6716	3.5159	
	Q21/A1	−0.0399 (−0.0917 to 0.0120)	0.1279	−0.3399	4.0761	
HALE in females	Q2/Y	0.7282 (0.2470 to 1.2095)	0.0040 **	0.4382	1.8087	0.5701
	Q4/Y	0.3824 (−0.1985 to 0.9634)	0.1906	0.2039	2.0697	
	Q5/Y	−0.3504 (−0.7263 to 0.0254)	0.0667	−0.2595	1.6705	
	Q10/Y	0.0457 (−0.0051 to 0.0965)	0.0765	0.2262	1.3641	
	Q13/Y	−0.1579 (−0.2795 to −0.0363)	0.0123 *	−0.3989	2.0355	
	Q18/A1	−0.1166 (−0.2066 to −0.0266)	0.0125 *	−0.3298	1.3982	
	Q22/Y	−0.0610 (−0.1200 to −0.0020)	0.0429 *	−0.2651	1.4157	

Question items (Q1–22) were included as candidate explanatory variables. The explanatory variables, which were finally included in the regression model and presented in this table, were selected from the candidate variables using the backward stepwise selection procedure with the criteria of *p* < 0.2. CI: confidence interval; B: partial regression coefficient; β: standardized partial regression coefficient; VIF: variance inflation factor; *R*^2^: coefficient of determination. * *p* < 0.05, ** *p* < 0.01 by *t*-test.

## Data Availability

Publicly available datasets were analyzed in this study. The data can be found here: [https://www.mhlw.go.jp/stf/seisakunitsuite/bunya/0000177221_00003.html].

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
