# Peer review of "The Significant Association between Health Examination Results and Population Health: A Cross-Sectional Ecological Study Using a Nation-Wide Health Checkup Database in Japan"

_ijerph, 2021, doi:10.3390/ijerph18020836_

Round 1

Reviewer 1 Report

Dear authors,

thank you for choosing the IJERPH journal for your submission. I think you have a very interesting and large database, including outcome measures that can be used to provide insights and recommendations on health promotion and prevention campaigns.

However, several important concerns arose from reading your manuscript. I hope my comments will help you in improving your research work.

In the introduction, you wrote that half of the eligible people participated in the check-up campaign. Who did participate and who did not? What are the characteristics of the two groups?

Did you consider if people who accessed health check-ups have more economic or educational resources? are they in a better socio-economic condition? are they more anxious people? or people living in urban areas? Could this represent a bias in your analysis and results' interpretation? In other words, is it possible that people are more healthy not because of health check-ups, but because they are more educated or richer or they live in a more urbanized area and this makes them more healthy and easier the access to check-ups? 

You mention this limitation in the discussions to explain the results (line 233) or to summarize limitations (line 391 ss). However, it is a key limitation... I would reconsider the structure of your hypothesis reconsidering the bias of participation... I suggest comparing the characteristics of your sample with characteristics of the whole reference population, extracting weights, and use them in your statistical analyses, moving from just descriptive statistics to inferential models. To do this, you have to use individual-level data. Why did you aggregate data to the prefecture-level? You lose a lot of data and information! 

You can and should perform more sophisticated analyses on your data. Please, add to simple correlations that you already run, almost some multilevel and multivariate regressions. In this way, you will quantify not only the direction and strength of the relationship between only two numeric variables, as you did, but you can also establish if and how each variable changes the outcomes (LE and HALE). This would make your results more interesting and robust.

ABSTRACT

I suggest avoiding acronyms.

line 26: please specify if positive association.

INTRODUCTION

line 33: please, define "health inequality". Does it refer not equal access to care, for instance? I would briefly describe the national health system of Japan (does it follow a berevidge or bismark model? how is it organized?).

METHODS

I anticipated the need to include more sophisticated analyses and to use data at the individual level. A simple correlation analysis is not enough to provide evidence and answer your research questions. It appears more similar to a report, while you have a great opportunity to provide insightful results from a very large and interesting database.

RESULTS

Scatter diagrams are not readable

DISCUSSIONs

They are too long. Please, shorten them by trying to extrapolate key results to discuss or by re-structuring the paragraph.

Author Response

[Reviewer’s comment] thank you for choosing the IJERPH journal for your submission. I think you have a very interesting and large database, including outcome measures that can be used to provide insights and recommendations on health promotion and prevention campaigns.

However, several important concerns arose from reading your manuscript. I hope my comments will help you in improving your research work.

[Response] We appreciate the evaluation on our manuscript and valuable suggestions from reviewer 1. We have made a thorough review of the points raised and revised the manuscript according to them. The changes in the revised manuscript were shown with blue text.

[Reviewer’s comment] In the introduction, you wrote that half of the eligible people participated in the check-up campaign. Who did participate and who did not? What are the characteristics of the two groups?

[Response] We appreciate the suggestion about the participants in the checkup. First, we added the following sentences to the Introduction for briefly explaining Japanese health care system. “The Japanese health care system is attributed to the Bismarck-type health insurance where all citizens are covered under one of the public health insurance programs. The programs for individuals aged 40 to 74 years are divided into 2 categories, i.e., the employment-based health insurance system wherein company employees and their family members are enrolled and the residence-based National Health Insurance system which is for people not eligible for the employment-based insurance system [8]. Public health insurers are obliged to provide a specific health checkup for the insured individuals aged 40 to 74 years to tackle lifestyle diseases as per government policy [9]. (P1, L41- P2, L47)” Then, according to the suggestion, we added the sentences regarding the participants to the Discussion (limitation). “The other limitation is related to the selection bias attributable to the relatively low participation rate (slightly higher than 50%) for the health checkup. Since participation is not mandatory for eligible adults (aged 40 to 74 years), the participation rate remains low and varies based on the insurers providing the checkup programs [54]. For example, participation rate was 75% for people insured by large companies’ insurance associations, one of Employee Health Insurance, and 37% for those insured by municipality-based National Health Insurance [54]. Thus, the participation rate is generally higher for employees than for self-employed persons, retirees, and non-working dependents. (P13, L354- 361)”

[Reviewer’s comment] Did you consider if people who accessed health check-ups have more economic or educational resources? are they in a better socio-economic condition? are they more anxious people? or people living in urban areas? Could this represent a bias in your analysis and results' interpretation? In other words, is it possible that people are more healthy not because of health check-ups, but because they are more educated or richer or they live in a more urbanized area and this makes them more healthy and easier the access to check-ups?

[Response] Thank you for the insightful suggestion regarding the limitation of this study. We added the following sentences as the limitation of this study. “This difference of properties between participants and non-participants could affect the findings. Indeed, the properties in terms of various factors, such as socio-economic condition, education level, financial status, and mental condition could affect population health outcomes. For example, a recent study which performed multiple regression analysis using Japanese population data aggregated by prefecture demonstrated that LE at 65 years of age in females was significantly affected by healthcare resource factors (beds per capita, doctors per capita, and medical expenses for the elderly) and an environmental factor (air pollution) [55]. Factorial analyses by including and controlling multiple factors, such as socio-economic-, mental condition-, and environment-associated variables must be focused on by future research. (P13, L361 – 370)”

[Reviewer’s comment] You mention this limitation in the discussions to explain the results (line 233) or to summarize limitations (line 391 ss). However, it is a key limitation... I would reconsider the structure of your hypothesis reconsidering the bias of participation... I suggest comparing the characteristics of your sample with characteristics of the whole reference population, extracting weights, and use them in your statistical analyses, moving from just descriptive statistics to inferential models. To do this, you have to use individual-level data. Why did you aggregate data to the prefecture-level? You lose a lot of data and information!

[Response] Thank you for the valuable suggestion regarding the limitation of our study. This study used the health checkup records in the fourth NDB Open Data Japan. This open data has already been aggregated at the prefecture level for privacy protection and cannot be analyzed at the individual level. As the reviewer suggests, this is the limitation of this study. We added the following sentences to the Discussion (limitation). “This study employed a cross-sectional ecological study design and used the health checkup data and the summary population health data collected in 2016 and 2015, respectively. One of the limitations of this study is that the health checkup data were aggregated prefecture-wise from the perspective of privacy protection. Hence, individual level analysis could not be conducted, limiting the findings to only indicate that health examination results are significantly associated with population health at the prefecture level. A study that uses individual data remains to be conducted to determine the factors associated with individual health.(P13, L348 - 354)”  

[Reviewer’s comment] You can and should perform more sophisticated analyses on your data. Please, add to simple correlations that you already run, almost some multilevel and multivariate regressions. In this way, you will quantify not only the direction and strength of the relationship between only two numeric variables, as you did, but you can also establish if and how each variable changes the outcomes (LE and HALE). This would make your results more interesting and robust.

[Response] Thank you for the insightful suggestion. According to the suggestion, we added multiple regression analysis using the summary population health outcomes as the objective variable and health checkup variables as explanatory variables. Results were summarized in tables 5 and 7.

[Reviewer’s comment] ABSTRACT  I suggest avoiding acronyms.

[Response] Thank you for the suggestion. We revised the Abstract by avoiding the use of abbreviations.

[Reviewer’s comment] line 26: please specify if positive association.

[Response] According to the suggestion, the sentence was revised as, “This study highlights that health examination results, including attitude toward improving health habits, are positively associated with population health. (P1, L26 – 27)”

[Reviewer’s comment] INTRODUCTION

line 33: please, define "health inequality". Does it refer not equal access to care, for instance? I would briefly describe the national health system of Japan (does it follow a berevidge or bismark model? how is it organized?).

[Response] Thank you for the suggestion. “Population health inequality” was used for “differences in LE”. For clarification, the sentence was changed to “Differences in LE among prefectures are increasing. (P1, L39 – 40)” According to the suggestion, we added the following sentences to the Introduction for briefly explaining Japanese health care system. “The Japanese health care system is attributed to the Bismarck-type health insurance where all citizens are covered under one of the public health insurance programs. The programs for individuals aged 40 to 74 years are divided into 2 categories, i.e., the employment-based health insurance system wherein company employees and their family members are enrolled and the residence-based National Health Insurance system which is for people not eligible for the employment-based insurance system [8]. Public health insurers are obliged to provide a specific health checkup for the insured individuals aged 40 to 74 years to tackle lifestyle diseases as per government policy [9]. (P1, L41 – P2, L47)”

[Reviewer’s comment] METHODS

I anticipated the need to include more sophisticated analyses and to use data at the individual level. A simple correlation analysis is not enough to provide evidence and answer your research questions. It appears more similar to a report, while you have a great opportunity to provide insightful results from a very large and interesting database.

[Response] Thank you for the suggestion regarding the analysis of individual level data. As the reviewer suggests, we appreciate that one of limitations of our study is that the Open Data has already been aggregated at the prefecture level from the perspective of privacy protection and cannot be analyzed at the individual level. We added the following sentences regarding the limitation and future studies to the Discussion (limitation). “This study employed a cross-sectional ecological study design and used the health checkup data and the summary population health data collected in 2016 and 2015, respectively. One of the limitations of this study is that the health checkup data were aggregated prefecture-wise from the perspective of privacy protection. Hence, individual level could not be conducted, limiting the findings to only indicate that health examination results are significantly associated with population health at the prefecture level. A study that uses individual data remains to be conducted to determine the factors associated with individual health. (P13, L348 - 354)”

[Reviewer’s comment] RESULTS  Scatter diagrams are not readable

[Response] Thank you for the suggestion. We added two tables 5 and 7 showing the multiple regression analysis results. In exchange for that, the scatter diagrams were omitted in the revised manuscript. We adjusted the number of figures and tables not to be too much, and followed another reviewer’s suggestion.

[Reviewer’s comment] DISCUSSIONs

They are too long. Please, shorten them by trying to extrapolate key results to discuss or by re-structuring the paragraph.

[Response] Thank you for your suggestion. We restructured the Discussion by focusing on the key results. As a result, the number of words decreased from 2588 to 1983.

Reviewer 2 Report

This study conducted a cross-sectional ecological design to explore the association between health examination results and summary population health among adult Japanese aged 40 -74.

Here are my comments:

  1. In the abstract (line 19), the authors may add abbreviations (LE and HALE) for life expectancy and healthy life expectancy because they used LE and HALE in the following sentences.

  1. It is good to standardize age to eliminate the age effect. However, LE and HALE might associate with economic factors (such as household income and urbanization) and health care factors (number of medical facilities and number of physicians). It may provide more information if these variables could be included.

  1. The sentence (line 114) “We age standardized the health check-up raw data by prefecture with the following formula:” might need to be checked.

  1. The authors might explain why they demonstrated scatter plots between certain health check-up variables and summary population health outcomes in figures 1 and 2 since the results had been revealed in the tables. For example, BMI is negatively associated with LE in both males and females. Why did the authors show a scatter plot among males only?

  1. The authors cited a paper and stated that “The association between BMI and relative mortality risk is reportedly J-shaped, i.e. high and extremely low BMI values are associated with increased all-cause mortality.” (line 228-229). However, in this study, the results only showed a negative correlation (-0.46 for males and -0.378 for females), which may not be fully supported by the reference they cited.

  1. In the discussion, the authors might need to discuss why their findings are consistent or inconsistent with the prior literature rather than just mention they are consistent or not. Additionally, some inconsistent findings were observed because of the study population selection. For example, In smoking habit-related variables (line286-287), authors stated “This result is not consistent with previous studies which showed that smoking habit was associated with reduction of HALE [40-42].“ These studies (references 40-42) targeted people who are either 65+ years old or However, this study included people aged 40-74 with a healthier status.

  1. In the limitation (line 384), the authors mentioned that “health check-up data and summary population health data (LE and HALE) were collected during roughly the same one year period.” However, in the method, they analyzed the health check-up variables from 2016 NDB Open Data Japan and the LE results were collected in 2015.

  1. The language is fine but can be improved throughout.

Author Response

[Reviewer’s comment] This study conducted a cross-sectional ecological design to explore the association between health examination results and summary population health among adult Japanese aged 40 -74.

[Response] We appreciate the evaluation on our manuscript and valuable suggestions from reviewer 2. We have made a thorough review of the points raised and revised the manuscript according to them. The changes in the revised manuscript were shown with blue text.

[Reviewer’s comment] 1. In the abstract (line 19), the authors may add abbreviations (LE and HALE) for life expectancy and healthy life expectancy because they used LE and HALE in the following sentences.

[Response] Thank you for the suggestion. Another reviewer also suggested that we should avoid acronyms in the Abstract. According to the suggestions from both reviewers, we revised the Abstract by avoiding the use of abbreviations.

[Reviewer’s comment] 2.  It is good to standardize age to eliminate the age effect. However, LE and HALE might associate with economic factors (such as household income and urbanization) and health care factors (number of medical facilities and number of physicians). It may provide more information if these variables could be included.

[Response] Thank you for the insightful suggestion. As the reviewer suggests, it is important to note that population health is possibly affected by factors other than health examination variables. We added the following sentences regarding the limitation and future studies to the Discussion (limitation) by citing a recently published literature which relates to reviewer's suggestion. “Indeed, the properties in terms of various factors, such as socio-economic condition, education level, financial status, and mental condition could affect population health outcomes. For example, a recent study which performed multiple regression analysis using Japanese population data aggregated by prefecture demonstrated that LE at 65 years of age in females was significantly affected by healthcare resource factors (beds per capita, doctors per capita, and medical expenses for the elderly) and an environmental factor (air pollution) [55]. Factorial analyses by including and controlling multiple factors, such as socio-economic-, mental condition-, and environment-associated variables must be focused on by future research. (P13, L362 - 370)”

[Reviewer’s comment] 3. The sentence (line 114) “We age standardized the health check-up raw data by prefecture with the following formula:” might need to be checked.

[Response] We appreciate the indication. We organized the sentences regarding the standardization of the raw data as follows. “The health checkup raw data were standardized by age using the following formula: Standardized data = (∑ age-specific raw data in a 5-year age group x standard population in that age group)/(total population in standard population); the Japanese population in 2015 [15] was used as the standard population. (P4, L115- 118)”

[Reviewer’s comment] 4. The authors might explain why they demonstrated scatter plots between certain health check-up variables and summary population health outcomes in figures 1 and 2 since the results had been revealed in the tables. For example, BMI is negatively associated with LE in both males and females. Why did the authors show a scatter plot among males only?

[Response] We appreciate the suggestion. According to the suggestion that the results had been revealed in the tables, scatter diagrams were omitted in the revised manuscript. We added tables 5 and 7 showing the multiple regression analysis results according to another reviewer’s suggestion.

[Reviewer’s comment] 5. The authors cited a paper and stated that “The association between BMI and relative mortality risk is reportedly J-shaped, i.e. high and extremely low BMI values are associated with increased all-cause mortality.” (line 228-229). However, in this study, the results only showed a negative correlation (-0.46 for males and -0.378 for females), which may not be fully supported by the reference they cited.

[Response] We appreciate the suggestion. We consider that individuals with extremely low BMI due to disease causing premature death tend to not participate in health checkups. This would be a possible reason why the linear negative relation between BMI and LE was obtained in this study. We revised the discussion as follows. “BMI and LE were negatively associated in case of both genders in the correlation study and regression analysis. The association between BMI and relative mortality risk was J-shaped. This implied that high and extremely low BMI values are associated with increased all-cause mortality. The increased mortality in those with extremely low BMI was partly due to the inclusion of individuals with diseases that cause weight loss and premature death and residual confounding by smoking [16]. The linear negative relation between BMI and LE in this study could happen, probably because individuals with diseases causing weight loss and premature death tend to not participate in health checkups. (P10, L232- 239)”

[Reviewer’s comment] 6. In the discussion, the authors might need to discuss why their findings are consistent or inconsistent with the prior literature rather than just mention they are consistent or not. Additionally, some inconsistent findings were observed because of the study population selection. For example, In smoking habit-related variables (line286-287), authors stated “This result is not consistent with previous studies which showed that smoking habit was associated with reduction of HALE [40-42].“ These studies (references 40-42) targeted people who are either 65+ years old or However, this study included people aged 40-74 with a healthier status.

[Response] We appreciate the valuable suggestions. As the result of careful consideration based on the suggestions, we deleted the following sentences because it is difficult to unequivocally define which is consistent or inconsistent to the previous evidence in general. “In general, the significant associations extracted are in consistent with the existing evidence, suggesting that the study design employed is valid to find health check-up variables which are significantly associated with summary population health. Careful interpretations are, however, necessary for some associations which are not in accordance with the leading hypotheses regarding the influence of the applicable variables on people's health.”

 As the reviewer suggests, the difference in age of subjects is a possible reason for the inconsistent finding. We added the following sentences in the revised manuscript. “This inconsistency could be due to different study designs. For instance, subjects aged 65 years or older were targeted in the previous study [36], while those aged 40-74 years were included in this study. (P11, L287- 289)” In addition, the results regarding cholesterols (LDL-C and HDL-C) in this study are considered inconsistent with their roles in the pathophysiologies of atherosclerosis. According to the suggestion, we mentioned why they are inconsistent as follows. “LDL-C and LE were positively correlated in males in the correlation study. Additionally, a negative association between HDL-C and HALE in females was observed in the correlation study and regression analysis. These results were inconsistent with their roles in the pathophysiology of atherosclerosis. Currently, LDL-C is considered a major causal factor for cholesterol transport to atherosclerotic lesions, whereas HDL-C performs a reverse transport of cholesterol to the liver [47]. Elevated LDL-C and decreased HDL-C levels are considered risk factors for atherosclerosis in cardiovascular disease [48]. (P12, L316- 322)”

[Reviewer’s comment] 7. In the limitation (line 384), the authors mentioned that “health check-up data and summary population health data (LE and HALE) were collected during roughly the same one year period.” However, in the method, they analyzed the health check-up variables from 2016 NDB Open Data Japan and the LE results were collected in 2015.

[Response] Thank you for the indication. We revised the sentence as follows. “This study employed a cross-sectional ecological study design and used the health checkup data and the summary population health data collected in 2016 and 2015, respectively. (P13, L348- 349)”

[Reviewer’s comment] 8. The language is fine but can be improved throughout.

[Response] According to the suggestion, the revised manuscript was proofread by English proofreading service (Cactus Communications, Website: https://www.editage.jp/).

Round 2

Reviewer 2 Report

The authors did a lot of works in improving the manuscript and addressed all the comments I provided in the last communication. In addition, I like the authors applied multiple linear regressions to examine the association between population health outcomes (LE/HALE) and laboratory tests/health habits. Here are my few suggestions:

  1. In lines 142, the authors stated “…as explanatory variables using the backward-forward stepwise method.” Did they use the bi-directional stepwise procedure to create regression models? They may need to clarify why they choose the procedure rather than one direction procedure (i.e. forward stepwise or backward stepwise). In addition, they might describe the inclusion and exclusion criteria regarding variables selection in section 2.5.
  2. Did the authors check the normality when creating linear models?
  3. The authors described variable measurements in the result section. I suggest the detailed information could be moved to the Method under section 2.5. For example, in lines 167-171 “BMI, fasting plasma glucose, HbA1c, systolic blood pressure, triglyceride, HDL-C, LDL-C, γ-GT, and hemoglobin were the explanatory variables in the analysis (BMI was used as body weight-related variable among BMI and abdominal circumscript). Systolic blood pressure was used as the blood pressure-related variable among systolic and diastolic blood pressure. γ-GT was used as liver enzyme-related variable among GOT, GPT, and γ-GT).

Author Response

[Reviewer’s comment]

The authors did a lot of works in improving the manuscript and addressed all the comments I provided in the last communication. In addition, I like the authors applied multiple linear regressions to examine the association between population health outcomes (LE/HALE) and laboratory tests/health habits. Here are my few suggestions:

[Response]

We appreciate the additional valuable suggestions from the reviewer. We revised the manuscript according to them. The changes in the revised manuscript were shown with blue text.

[Reviewer’s comment]

  1. In lines 142, the authors stated “…as explanatory variables using the backward-forward stepwise method.” Did they use the bi-directional stepwise procedure to create regression models? They may need to clarify why they choose the procedure rather than one direction procedure (i.e. forward stepwise or backward stepwise). In addition, they might describe the inclusion and exclusion criteria regarding variables selection in section 2.5.

[Response]

As the result of careful consideration based on the suggestion, we used the backward stepwise selection method in the revised manuscript. As the reviewer suggests, there is no specific reason to use bi-directional stepwise method in the present study. The analysis results whose values have changed due to this change of method were shown with blue text in the revised manuscript. The criteria used for variables selection was p < 0.2. We added the following description in in section 2.5. according to the suggestions. “using the backward stepwise selection method with the criteria for selection set at p < 0.2. (P5, L141- 142)”.

[Reviewer’s comment]

  1. Did the authors check the normality when creating linear models?

[Response]

We appreciate the suggestion. We added the sentences regarding the confirmation of normality in the present study. “Quantile-Quantile (Q-Q) plot was used to verify the normality assumption of the residuals in the regression analysis. (P6, L149- 150)”. “Normality of residual distribution in each regression analysis was met as assessed by Q-Q plot. (P7, L178- 179)”. “Normality of residual distribution in each regression analysis was met as assessed by Q-Q plot. (P9, L220)”.

[Reviewer’s comment]

  1. The authors described variable measurements in the result section. I suggest the detailed information could be moved to the Method under section 2.5. For example, in lines 167-171 “BMI, fasting plasma glucose, HbA1c, systolic blood pressure, triglyceride, HDL-C, LDL-C, γ-GT, and hemoglobin were the explanatory variables in the analysis (BMI was used as body weight-related variable among BMI and abdominal circumscript). Systolic blood pressure was used as the blood pressure-related variable among systolic and diastolic blood pressure. γ-GT was used as liver enzyme-related variable among GOT, GPT, and γ-GT).”

[Response]

Thank you for the suggestion. According to the suggestion, we moved the corresponding sentences to the Method (section 2.5.) as follows. “In the regression analysis using the laboratory tests, BMI, fasting plasma glucose, HbA1c, systolic blood pressure, triglyceride, HDL-C, LDL-C, γ-GT, and hemoglobin were the explanatory variables (BMI was used as body weight-related variable among BMI and abdominal circumscript. Systolic blood pressure was used as the blood pressure-related variable among systolic and diastolic blood pressure. γ-GT was used as liver enzyme-related variable among GOT, GPT, and γ-GT). (P5, L142- 146) “. “In the regression analysis using the answer to the questions regarding health habits, all question items (Q1-22) were included as explanatory variables. Only answer option 1 was the target of analysis for question items with 3 or more answering options. (P5, L146 – P6, L149) “.
